# An Association between Alexithymia and the Characteristics of Sport Practice: A Multicenter, Cross-Sectional Study

**DOI:** 10.3390/healthcare10030432

**Published:** 2022-02-25

**Authors:** Catarina Proença Lopes, Edem Allado, Mathias Poussel, Aghilès Hamroun, Aziz Essadek, Eliane Albuisson, Bruno Chenuel

**Affiliations:** 1Development, Adaptation and Disadvantage, Cardiorespiratory Regulations and Motor Control (EA 3450 DevAH), University of Lorraine, 54000 Nancy, France; e.allado@chru-nancy.fr (E.A.); m.poussel@chru-nancy.fr (M.P.); b.chenuel@chru-nancy.fr (B.C.); 2Center of Sports Medicine and Adapted Physical Activity, CHRU-Nancy, University of Lorraine, 54000 Nancy, France; 3Nephrology Department, Regional and University Hospital Center of Lille, Lille University, 59000 Lille, France; aghiles.hamroun@chru-lille.fr; 4INTERPSY, University of Lorraine, 54000 Nancy, France; aziz.essadek@univ-lorraine.fr; 5InstitutElie-Cartan de Lorraine, CNRS, Université de Lorraine, 54000 Nancy, France; e.albuisson@chru-nancy.fr

**Keywords:** alexithymia, sport, competition, training, recreational practice

## Abstract

Background: This was a multicenter, cross-sectional study which aimed to investigate the relationship between the characteristics of sport practice (weekly training duration, level of practice) and alexithymia in adults who were officially licensed at a sports club. Methods: From a sample of sports club licensed adults, 188 participants were included. The participants completed computerized questionnaires on anthropometric data and characteristics of sport practice (level and weekly time spent on sport practice) as well as alexithymia (TAS 20), depression (BDI-13) and anxiety traits (STAI-Y form B). Results: In this sample, 91 (48.4%) and 97 (51.6%) athletes engaged in recreational and competitive sport practice, respectively. We observed a prevalence of 31.9% for alexithymia. Moreover, alexithymics were more involved in competitive than recreational practice (40.2% versus 23.1%, respectively; *p* = 0.019) and they were less anxious (63.9% versus 80.2%, respectively; *p* = 0.010). Finally, alexithymia was significantly more pronounced than non-alexithymia among sports competition practitioners (OR: 3.57 (95 CI [1.26–10.08]; *p* = 0.016) and we observed less alexithymia in team sports practice than confrontation sports (OR: 0.20 (95 CI [0.05–0.78]; *p* = 0.020). Conclusions: Alexithymic athletes were more involved in competition than recreational sports compared to non-alexithymic subjects, whilst there were more alexithymic athletes in confrontation sports than in team sports.

## 1. Introduction

Recently, alexithymia has been widely described as a common personality trait, with a prevalence rate varying from 8 to 20% [1]. It is mainly characterized by a deficit in: the ability to recognize and express emotions; use of concrete speech and thoughts related to external events; and a paucity of notional thought processes [2]. Since personality plays a substantial role in performance—more precisely accounting for up to 45% of the variance in sports performance—the relationship between alexithymia and sport needs to be clarified [3]. It has been only rarely studied in sports, and it has often been portrayed as deleterious to performance [4]. Nevertheless, the athlete’s ability to manage their emotions during a competition is considered of crucial importance and its development is encouraged throughout his or her career [5,6]. Otherwise, as is the case with alexithymics, athletes are likely to be considered less resilient or less mentally resistant [7]. Paradoxically, however, the stressful nature of the competitive environment could become an advantage for alexithymics, allowing alexithymic individuals to be drawn to competition settings. For this reason, alexithymia has been mainly studied in the field of high-risk sports in which athletes are able to favor their need for regulating feelings and anxiety over the difficulty of developing their emotions [8,9,10,11,12]. In high-risk domains, alexithymics are in a better position to push themselves to their physical limits and thus alexithymic prevalence is markedly higher than in the general population, reaching 30% in scuba-diving, climbing, skydiving, sea-rowing, mountaineering, and rowing [8,9,10,11,12]. Theories suggest that the high-risk environment would provide benefits of regulating emotions and thus experience mastering anxiety, which can be particularly beneficial to alexithymic sports practitioners [9,11,13,14,15].

It is important to notice that alexithymia is particularly associated with anxiety and depression in the general population [16]; each influences the development of the others. To our knowledge, the alexithymia/anxiety link in sport has been only poorly studied as it has in depression. Some studies have shown that alexithymic athletes would be more anxious than the non-alexithymic [8,10]. These levels of depression and alexithymia are predictors of competitive anxiety [17]. Moreover, various studies have shown that alexithymia would be a potential moderator of anxiety fluctuations since there is greater anxiety in alexithymic athletes than in the non-alexithymic [8,10]. Nevertheless, the link has yet to be determined. Indeed, several transverse and longitudinal empirical studies have produced contradictory results. We aimed to evaluate the presence or absence of anxiety and depression in different sports and also to study a potential relationship with the level of practice.

The positive relationship between the extent of sports training and the level of alexithymia, has also been discussed, both positively and negatively [18,19]. Moreover, a clear link between alexithymia and dependence on exercise has already been demonstrated, with higher alexithymia predicting more severe exercise dependence [20]. Some addiction to risky sports can also be observed in alexithymic sportsmen. This is explained by the short-term emotional benefits (especially in terms of anxiety) brought by this sport causing a continuing search for regulation through its practice. [13,21]. It will be interesting to see if it is only present in high-risk sports or if competition conditions provide the necessary stimulus.

Our intention, therefore, was to evaluate the degree of alexithymia in different types of sports (confrontational, collective, individual) and to study the relationship between alexithymia prevalence and the characteristics of sport practice (amount and level of training, recreational versus competitive) in individuals who were officially licensed with a sports club.

## 2. Materials and Methods

### 2.1. Participants

This cross-sectional study was performed from January to May 2017 among a sample of sports practitioners currently licensed at a sports club and who had given their consent. The inclusion was made possible by contacting various sports clubs in France who distributed the questionnaires and informed consent forms in computerized file format to their members by e-mail. The inclusion criteria were age greater than 18 years, active sports practitioners, and an understanding of the written and the oral French language. Only a diagnosed psychiatric pathology was considered a criterion for non-inclusion.

### 2.2. Procedure

To assess sports practice characteristics more precisely, data were collected regarding the level and the weekly time spent on sports practice. Sports were pooled according to the following categories:-Confrontation sports (i.e., against an opponent): racket sports (tennis, badminton), fighting sports, martial arts.-Team sports: football, rugby, handball, volleyball, ice hockey.-Individual sports: swimming, climbing, golf, diving, track and field, archery, skiing, crossfit.-Endurance sports: cycling, running, rowing, trail running, ultra-trail.-Stretching/dance sports: Zumba, fitness, gymnastics, yoga, dance.

The age and sex of participants were also included. Study intervention comprised three questionnaires to assess three different psychological aspects. We chose these following instruments because they are the most used in the field and they have good psychometrics properties. Moreover, they were all self-administered questionnaires that were quick (5–10 min) and easy to complete, which was important because we were not in the presence of the subject during the administration of the questionnaire.

1. Toronto Alexithymia Scale (TAS-20) [2] is composed of three subscales which are combined to produce a total alexithymia score (high scores equate to high alexithymia). A score between 45 and 55 defines a ‘borderline’ status for alexithymia and a score greater than or equal to 56 is a sign of the proven presence of alexithymia. The three-factor structure are: difficulty identifying feelings (DIF) with 7 items, difficulty describing feelings (DDF) with 5 items, and externally oriented thinking (EOT) with 8 items. Both the reliability and the validity of the TAS-20 have been amply demonstrated. Internal consistency was good for the total TAS-20 scale (alpha = 0.79), the DIF (alpha = 0.77), and the DDF (alpha = 0.72) subscales, but the internal consistency of the EOT subscale was only moderate (alpha = 0.61).

2. Beck’s Short Depression Inventory (BDI-13) [22] measures with 13 items the intensity of the depressive state and varies from 0 to 39: a slight depression is graded from 4, depression is moderate if greater than 8, and it is severe if greater than 16. The internal consistency was described as approximately 0.9 and the retest reliability ranged from 0.73 to 0.96.

3. Spielberger’s State–Trait anxiety questionnaire (STAI-Y) [23] quantifies the level of anxiety. It differentiates between a state of anxiety, (anxiety as an emotional state related to a particular situation (feelings of nervousness, worry felt at a specific moment) and trait anxiety (assessment of anxiety as a personality trait (feelings of apprehension, tension, nervousness and worry that the subject usually feels) splitting into the S-Anxiety scale and T-Anxiety scale, each having 20 items of a 1–4 scale and score ranges from 20 to 80 each. For each scale, anxiety above 56 is considered high and very high if it is above 65. We used the T-Anxiety scale (Y-B form) only. It demonstrates high internal consistency, with alpha coefficients of 0.86 and 0.81.

All of the questionnaires have been translated and validated in French [24,25].

### 2.3. Data Analysis

Both descriptive and comparative analyses were conducted according to the nature and the distribution of the variables. Qualitative variables are described with frequencies and percentages; quantitative variables are reported as mean ± standard deviation (SD). The chi-square test, with Fisher’s exact test if necessary, was used for the ordinal or nominal data analysis. The student t test was used for the comparisons of quantitative variables with normal distribution. Non-parametric tests were used for non-normal distribution variables and multinomial logistic regression analyses were performed to describe and study the association between a sport’s practice and alexithymia. Analyses were performed using IBM SPSS Statistics V.23, and *p* values < 0.05 were considered statistically significant.

### 2.4. Legal Obligation

All participants received clear and fair information to enable them to provide informed consent. The study is consistent with the ethical principles of the Helsinki declaration.

## 3. Results

We evaluated the degree of alexithymia in different types of sports and studied the relationship between alexithymia prevalence and the characteristics of sport practice, taking into consideration the probable influence of anxiety and depression.

### 3.1. Descriptive Characteristics of Participants and Alexithymia Prevalence

A total of 191 responses were received, but three incomplete questionnaires were excluded. Thus, 188 participants were included, with an average age of 28.9 (11.2) years and a majority being men (54.3%). The most common types of sports were individual sports (34.5%). Overall, we observed a prevalence of 31.9% for alexithymia in our sample of sports licensed adults (Table 1).

### 3.2. Association between Level of Sports Practice and Alexithymia

Among the sample of participants, 97 (51.6%) were engaged in competition practice. The average age of those who practiced in competition was significantly younger than that of those engaged in recreational sports (33.5 years versus 24.7; *p* ≤ 0.001). We found a greater prevalence of alexithymia in participants involved in competitive practice than those in recreational practice (40.2% versus 23.1%, respectively; *p* = 0.019). We observed less anxiety in the competition group (63.9% versus 80.2%, respectively; *p* = 0.010). No other significant association was identified regarding depression (Table 1).

Sixty-six (35.1%) participants used to practice less than 5 h/week and 122 (64.9%) reported a sports practice of more than 5 h/week. A significant positive relationship was found between the number of hours practiced and the competitive sport practice, since 65.6% of athletes who engaged in more than 5 h of sport were also competitors while only 25.8% of participants who used to practice recreationally performed at competition level (*p* < 0.001). No significant difference was identified regarding depression, anxiety, and alexithymia (Table A1, in Appendix A).

### 3.3. Relationship between Alexithymia, Demographic Characteristics, and Sports Practice

In this sample, alexithymics were younger than borderline and non-alexithymia participants (25.9 versus 29.5 and 32.0 years, respectively; *p* = 0.019). No difference in terms of sex or length of practice was observed. There was no significant difference in anxiety depending on the type of sport practiced. However, we observed a significant difference in the number of depressive alexithymic athletes compared to those who were borderline and non-alexithymic (30.0% versus 6.0% and 4.4% respectively; *p* < 0.001) (Table 2).

Moreover, in multivariate analyses, there was more alexithymia at competition level than in recreational practice (OR: 3.57 (95 CI [1.26–10.08]; *p* = 0.016). We observed less alexithymia in team sports practice than confrontation sports (OR: 0.20 (95 CI [0.05–0.78]; *p* = 0.020) (Table A2, in Appendix A). The same results were found in borderline status for team sports (OR: 0.24 (95 CI [0.07–0.80]; *p* = 0.020) and endurance sports (OR: 0.48 (95 CI [0.13–0.99]; *p* = 0.048) (Table A3, in Appendix A).

## 4. Discussion

Our study showed more alexithymia in competition sports practitioners than recreational practitioners. In particular, there were a greater number of alexithymic and borderline alexithymic athletes in competitive sports than in team sports.

The prevalence of alexithymic and borderline individuals in the population studied here is extremely high. We found a prevalence of alexithymia in competitors that was twice as high as in recreational sports. Although we observed that prevalence for recreational practice was the same as that in the general population (between 17% and 23%) the prevalence in competitive practice exceeded that found in professional/experienced athletes (between 9% and 33%) [10,13,19,26]. It also supports the idea that alexithymia can be observed outside the field of extreme sports. People with alexithymia may therefore have a particular appetite for physical activity or find a way to satisfy/express a need for it. Through sport they can overcome difficult challenges using active, bodily expression or even by using a concrete non-introspective way of thinking [27].

We also found a link between alexithymia and depression, similar to that reported in the literature for the general population [28]. As said, to our knowledge, the alexithymia/depression link in sport has been only poorly studied and research works can be contradictory. Here, our results are in line with those of Aston et al., regarding both retired and active hockey players. In fact, more pronounced alexithymia was associated with greater depressive symptoms [29]. We know now that depression acts as a strong moderator between alexithymic features and psychopathology. Indeed, alexithymia can, for example, facilitate the maintenance of depression [30].

Other researchers have reported that levels of depression and alexithymia are predictors of competitive anxiety [17]. As a result, the higher the levels of depression and alexithymia, the more anxious an athlete would be approaching a competition [17]. Moreover, various studies have shown that alexithymia would be a potential moderator of anxiety fluctuations since there is greater anxiety in alexithymic athletes than in the non-alexithymic [8,10]. In high-risk sports, theories suggest that this environment would provide benefits of regulating emotions and thus experiencing and mastering anxiety, which can be particularly beneficial [9,11,13,14,15]. Here we observed less anxiety in the competition group and a significant positive relationship was found between the number of hours practiced and the competitive sport practice. We can suppose that intensive training allows alexithymic practitioners to regulate their emotions and more particularly their anxiety.

This study also showed that alexithymia was predominant in team sports relative to confrontation sports. To our knowledge, it is the first time this observation has been highlighted. We can hypothesize that this could be induced by a preferential orientation of alexithymic athletes towards these sports or because this type of sport develops alexithymic traits in practitioners.

The practice of sport in competition implies a personal investment as it impacts daily life and lifestyles, requires hours of training per day, necessitates the learning of pain management, and tests the body’s limits. The body is, in fact, a working tool at the center of the athlete’s life, on which attention is focused since it can convey sensations and expression other than through words. Thus, the link is made with extreme/risk sports. In fact, “Individuals with alexithymia tend to avoid unwanted thoughts and feelings, they may be more likely to engage in risky behavior as an emotion-regulation strategy” [12].

Our results raise a fundamental question: are alexithymic people attracted to sport—and to practice in competition—or can training be the cause of the development or increase of alexithymia? A similar question is shared by other authors, such as Barlow et al., (2015), regarding the emotion regulation motives that might be associated with risk taking in sport and exercise environments more globally. We give some thought as follows. Firstly, the influence of the family environment and then development of alexithymia could be a process of adaptation to sport training.

A study by Top and Akil [31] showed that the level of alexithymia and social competence of families can affect children’s orientation to sports. In fact, the levels of alexithymia in children of families engaged in sports were significantly higher than in children who had no such involvement. We could thus speculate that the children of someone with alexithymia might progress to competitive sports. This would explain the higher number of alexithymic athletes in competitive practice in our study.

We might question the description of the development of alexithymia as a process of adaptation to its environment (here, professional sports training). Thus, facing an event triggering an emotional experience, two possibilities are available to the athlete: repression or expression of emotion. Neither strategy distinguishes superior effectiveness as the advance of research in extreme situations demonstrates [32]. Nevertheless, the effectiveness of one of the two strategies depends on the flexibility of response and its suitability for a specific situation [33]. The notion of subjectivity is therefore important. Although the athlete is not alexithymic before starting intensive training as part of his preparation for a championship, it is his experience and his personal background that could influence whether alexithymia develops or not. This could be an interesting area to develop in future research.

The main limitation of the study is a selection bias related to the voluntary participation of responders. We had few participants to distribute between the various categories and therefore had an insufficient number for the creation of homogeneous and robust groups. It therefore seems appropriate to undertake further research using a larger sample.

In addition, this research allows us to highlight the presence of alexithymia in different sports other than extreme sports, which to our knowledge, are those most frequently covered by sport/alexithymia studies. Furthermore, the recruitment of athletes was carried out in several centers in France, and the questionnaire was validated for measurement of alexithymia as well as for depression and anxiety. Nevertheless, we have suggested an original idea regarding the connection between alexithymia and a possible benefit for sports performance.

## 5. Conclusions

In conclusion, this study is to our knowledge, the first to observe that alexithymia was predominant in team sports and in confrontation sports, but it highlights almost a higher prevalence of alexithymia in competitive athletes who need to train intensively to perform. So, alexithymia is present in many sports, other than high-risk sports. However, the reason for this observation, whether training, environment or stress, remains unknown and needs to be clarified by further studies.

## Figures and Tables

**Table 1 healthcare-10-00432-t001:** Baseline demographic, mental and sports characteristics (*n* = 188).

Athletic Group	Total (*n* = 188)	Recreational (*n* = 91)	Competition (*n* = 97)	*p*-Value
Age	28.9 (11.2)	33.5 (12.3)	24.7 (9.0)	<0.001
Sex (women)	86 (45.7)	45 (49.5)	56 (57.7)	0.380
Amount of sport (per week)				
<5 h	66 (35.1)	49 (53.8)	17 (17.5)	<0.001
>5 h	122 (64.9)	42 (46.2)	80 (82.5)	
Type of sport				
Confrontation sports	18 (9.6)	4 (4.4)	14 (14.4)	<0.001
Stretching/dance sports	12 (6.4)	11 (12.1)	1 (1.0)	
Team sports	38 (20.2)	4 (4.4)	34 (35.1)	
Endurance sports	55 (29.3)	42 (46.2)	13 (13.4)	
Individual sports	65 (34.5)	30 (33.0)	35 (36.1)	
Depression (BDI)				
No	107 (56.9)	57 (62.6)	50 (51.5)	0.299
Mild	56 (29.8)	23 (25.3)	33 (34.0)	
Medium and severe	25 (13.3)	11 (12.1)	14 (14.4)	
Anxiety (STAI)				
Severe or more (>56)	135 (71.8)	73 (80.2)	62 (63.9)	0.010
Alexithymia (TAS 20)				
No (<45)	45 (23.9)	28 (30.8)	17 (17.5)	0.019

Data are presented as the mean (±SD) for age and as *n* (%) for other variables. The chi-square test was used for analysis of sex, type of sport, amount of sport. The student t test was used for age. Mann–Whitney test was used for analysis of anxiety and depression. Abbreviations: BDI = Beck’s Short Depression Inventory; STAI = Spielberger’s State–Trait Anxiety Inventory; TAS = Toronto Alexithymia Scale.

**Table 2 healthcare-10-00432-t002:** Relationships between alexithymic participants and depression, anxiety, sport characteristics.

	Non Alexithymic (*n* = 45)	Borderline (*n* = 83)	Alexithymic (*n* = 60)	*p*-Value
Age	32.0 (12.4)	29.5 (11.5)	25.9 (9.4)	0.019
Sex (women)	24 (53.3)	49 (59.0)	29 (48.3)	0.443
Amount of sport (per week)				
<5 h	18 (40.0)	30 (36.1)	18 (30.0)	0.549
>5 h	27 (60.0)	53 (63.9)	42 (70.0)	
Level of practice				
Recreational	28 (62.2)	42 (50.6)	21 (35.0)	0.019
Competition	17 (37.7)	41 (49.4)	39 (65.0)	
Type of sport				
Confrontation sports	3 (6.7)	7 (8.4)	8 (13.3)	0.255
Stretching/dance sports	3 (6.7)	6 (7.2)	3 (5.0)	
Team sports	11 (24.4)	14 (16.9)	13 (21.7)	
Endurance sports	19 (42.2)	21 (25.3)	15 (25.0)	
Individual sports	9 (20.0)	35 (42.2)	21 (35.0)	
Depression (BDI)				
No	30 (66.7)	53 (63.9)	24 (40.0)	<0.001
Mild	13 (28.9)	25 (30.1)	18 (30.0)	
Medium and severe	2 (4.4)	5 (6.0)	18 (30.0)	
Anxiety (STAI)				
severe or more (>56)	34 (75.6)	59 (71.1)	42 (70.0)	0.806

Data are presented as the mean (±SD) for age and as *n* (%) for other variables. The chi-square test was used for analysis of sex, type of sport, amount of sport. The student t test was used for age. Mann–Whitney test was used for analysis of anxiety and depression. Abbreviations: BDI = Beck’s Short Depression Inventory; STAI = Spielberger’s State–Trait Anxiety Inventory; TAS = Toronto Alexithymia Scale.

## Data Availability

Not applicable.

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
