# Peer review of "An Association between Alexithymia and the Characteristics of Sport Practice: A Multicenter, Cross-Sectional Study"

_healthcare, 2022, doi:10.3390/healthcare10030432_

Round 1
Reviewer 1 Report
This was a multicenter, cross-sectional study which aimed to investigate the relationship between characteristics of sport practice (weekly training duration, level of practice) and alexithymia in adults who were officially licensed at a sports club. From a sample of sports club licensed adults, 188 participants were included. The participants filled out computeized questionnaires on anthropometric data and characteristics of sport practice (level and weekly time spent on sport practice) as well as alexithymia (TAS 20), depression (BDI-13) and anxiety traits (STAI-Y form B).
The authors concluded that Alexithymic athletes were more involved in competition than recreational sports com- 29 pared to non-alexithymic subjects, whilst there were more alexithymic athletes in confrontation 30 sports than in team sports.
This is a very good piece.
The design is clear.
The article is well written.
The rationale flows well.
Some minor suggestions:
- Introduce a table with acronyms
- Introduce with a few sentences the paragraphs of the results
- Better develop the conclusions.
Reviewer 2 Report
First of all, I would like to congratulate the authors for such a commendable job. The research presented is extremely interesting and provides great possibilities for a broad prospective in the near future.
However, I consider that there are small issues that need to be improved with a view to the final publication of this research project.
Firstly, the introduction is too concise and develops the most important parts too briefly. It only dedicates 4 lines (53-56) to the previous similar articles, and it does so in a very perfunctory way.
Second, it also describes the evaluation instruments too lightly in the method section. It does not explain why they have selected these questionnaires and not others, nor does it show important information about them such as the reliability coefficient or their psychometric properties.
Regarding the discussion, numerous new scientific articles are cited and compared in that section with respect to the introduction, when it would be advisable to have exposed some of these reference works in the introduction to better present the state of the issue in which the article is framed.
Finally, both the results and their implications are extremely interesting and important for the scientific knowledge of sports practice. However, he barely dedicates three lines to it (223-226), so it should be significantly expanded and enriched.
